# Immune-Mediated Inflammation in Vulnerable Atherosclerotic Plaques

**DOI:** 10.3390/molecules24173072

**Published:** 2019-08-23

**Authors:** Harald Mangge, Gunter Almer

**Affiliations:** Clinical Institute of Medical and Chemical Laboratory Diagnostics, Medical University of Graz, A-8036 Graz, Austria

**Keywords:** atherosclerosis, cardiovascular disease, immune activation, inflammation

## Abstract

Atherosclerosis is a chronic long-lasting vascular disease leading to myocardial infarction and stroke. Vulnerable atherosclerotic (AS) plaques are responsible for these life-threatening clinical endpoints. To more successfully work against atherosclerosis, improvements in early diagnosis and treatment of AS plaque lesions are required. Vulnerable AS plaques are frequently undetectable by conventional imaging because they are non-stenotic. Although blood biomarkers like lipids, C-reactive protein, interleukin-6, troponins, and natriuretic peptides are in pathological ranges, these markers are insufficient in detecting the critical perpetuation of AS anteceding endpoints. Thus, chances to treat the patient in a preventive way are wasted. It is now time to solve this dilemma because clear results indicate a benefit of anti-inflammatory therapy per se without modification of blood lipids (CANTOS Trial, NCT01327846). This fact identifies modulation of immune-mediated inflammation as a new promising point of action for the eradication of fatal atherosclerotic endpoints.

## 1. Introduction

Myocardial infarction and stroke are still the leading causes of death worldwide. Atherosclerosis (AS) a sub-acute immune-mediated inflammation around lipid accumulations of the vascular wall is responsible for this burden. A main feature of AS is infiltration of the vessel wall by monocytes and T-cells. These cells interact with one another and with the arterial wall cells [1,2,3]. Thus, a chronic inflammatory process causes formation, progression, and rupture of lesions called AS-plaques [1,4,5,6]. A long-term activation of the innate immune system plays a central role in the pathologic cascade leading to symptomatic AS lesions [7].

Effective identification methods and techniques to follow the development of AS-plaques before clinical events occur are missing. This is caused not only by the lacking performance of blood biomarkers and imaging techniques but also by the availability of specific molecules for targeted recognition [1,8,9]. Thus, in most cases, vulnerable AS lesions become detectable too late by either incidence of myocardial infarction or stroke or the effect of arterial stenosis on organ perfusion. As vulnerable AS plaques are frequently non-stenotic, preclinical identification is not possible [1,9,10].

## 2. Special Features of the Vulnerable Atherosclerotic Plaque

Vulnerable AS plaques are unstable structures changing dynamically with time. They occur more frequently in regions of non-uniform shear stress, e.g., around bifurcations of the carotid or coronary arteries, and they are very often non-stenotic because they expand to the perivascular regions. A specific three-dimensional shape and cellular composition are key features of vulnerability [9,11]. Vulnerable plaques have a fragile, thin fibrous cap, an expanded lipid core, intraplaque hemorrhage, immune activation, increased production of pro-inflammatory mediators (cytokines, chemokines, metalloproteinases), and strong activity of certain macrophage subtypes [12,13,14]. Adventitial neovascularisation, low calcification, and fibrosis are further present. The true role of plaque calcification [15] remains a matter of debate. Two types of calcification exist—macro-calcification and micro-calcification [16]. Especially, micro-calcifications located within the fibrous cap are critical for plaque destabilization [16], whereas macro-calcification have been connected with stronger stability of AS plaques [16]. The pro-inflammatory micro-calcification associates with M1 polarization of macrophages, and macro-calcification associates with anti-inflammatory M2 polarization [17]. Thus, the classic view supposing calcification as a passive one-way process has been left, and a new sight suggests that calcification involves passive and active processes [16]. The passive calcification is more a biochemical process, whereas a cell-mediated process causes active calcification via cell death of macrophages and smooth muscle cells matrix vesicles [16].

Macrophage apoptosis and the ability of macrophages to clean up dead cells, a process called efferocytosis, are further crucial determinants of atherosclerotic lesion progression [18]. Efferocytosis comprises an interaction of receptors, bridging molecules, and apoptotic cell ligands [18]. Unstable AS lesions have a large necrotic core with ineffective efferocytosis [19]. Endoplasmatic reticulum (ER) stress, inflammatory activation, and lipid peroxidation contribute to “deficient” apoptosis and efferocytosis [18,19]. To repair the inefficient efferocytosis machinery may stabilize unstable AS lesions. Recent studies identified the CD47 “do not eat me” signal as a potential therapeutic target in this context [20].

Molecular imaging can help to improve the early diagnosis of plaque instability. Nanoparticles (NPs) synthesized from sodium fluoride, polyethylene glycol, or iron oxide have been examined for this [21,22,23,24]. Contrast agents consisting of iron oxide enable iron labelling in macrophages for MRI-based detection [22]. A limitation of molecular imaging is that it takes a long delay (2–24h) between contrast injection and post contrast imaging. Thus, this imaging is more complex compared to computed tomography, which lacks specificity.

## 3. Essential Mechanisms Promoting Vulnerability

### 3.1. Inflammation—The Monocyte/Macrophage System

Immune-mediated inflammation is a main driver of AS, and monocytes/macrophages are principal offenders [25,26]. As main antigen presenters and scavengers of the innate immune system in regions of inflammatory activation, macrophages become lipid-laden foam cells and operate the inflammatory milieu in the atherosclerotic vessel wall by secretion of cytokines and chemokines and their interaction with other immune cells in the plaque [27]. Destabilization of the atherosclerotic plaque associates with increased matrix metalloproteinase and cytokine production of macrophages and other cells including damage of the plaque’s extracellular matrix [28,29].

#### 3.1.1. Transdifferentiation

The origin of plaque macrophages is currently a matter of debate. They may enter from the blood circulation or from the perivascular site, and they develop from other cell types of the inflamed tissue in a process called transdifferentiation. Allahverdian et al. suggest that vascular smooth muscle cells (VSMCs) can gain functions of a macrophage, like lipid uptake [30] or phenotypic conversion toward monocyte descent [31,32]. Residing before disease initiation in the aortic media region, VSMCs may mobilize into the intima during atherosclerosis [32,33], where they lose their markers of contractile smooth muscle cells. Nevertheless, despite phenotypic resemblances, it is unclear whether such transdifferentiated cells achieve the same functions as classical plaque macrophages. Other authors doubted the assumption that every monocyte turns into a macrophage [28,29]. They suggest that monocytes entering inflamed tissue act more as short-dated antigen-presenting cells, which move to lymph nodes without significant differentiation. [34]. Presumably, as atherosclerosis progresses, they re-enter vascular lesions and proliferate locally [35]. On the other hand, AS plaques may be colonized with macrophages by regeneration and proliferation of tissue-resident cells. An observation that supports this hypothesis is that proliferation signals develop particularly in fatty streaks that are rich in macrophages [36,37] and where co-localization with foam cells occurs [38].

#### 3.1.2. Polarization

Monocytes and macrophages are the main effector cells in atherosclerosis. Nevertheless, it is still unclear how these cells combine to mark progression of human cardiovascular disease. Although a complex taxonomy of different macrophage subtypes exists so far [39,40], a rather confusing nomenclature of subsets arising during the so called polarization process does not give a reliable basis for clinically usable biomarkers of imminent atherosclerotic decompensation. Basically, the M1 macrophage subset acts proinflammatory, and the M2 subset acts anti-inflammatory. The transcriptome of M1 macrophages supports a Th1, that of M2 macrophages a Th2 adaptive immune response. Macrophage subsets change permanently in the atherosclerotic plaque at different stages. Taken together, complex interactions between and influencing factors upon macrophage types contribute significantly to the direction of pathologic development: stabilization → destabilization → exacerbation, or labile chronification, respectively. Cytokines (e.g., GM-CSF, M-CSF, LPS/INFγ, IL-4, IL-10, IL-12, IL-13, IL-17), fatty acids, immune complexes, metalloproteinases, bleeding processes, and lipid peroxidation are all important influential factors [41]. Even disturbed circadian rhythms are a matter of debate [42]. Accordingly, melatonin stabilized rupture-prone vulnerable plaques by suppression of M1 polarization [42]. Moreover, evidence exists that the melatonin-RORα (nuclear receptor retinoid acid receptor-related orphan receptor-α) axis can act as an endogenous protective signaling pathway in the vasculature [42]. Nevertheless, a limitation of many of these observations is that they come from in vitro experiments or animal models. A transfer to the human situation must keep attention. Concerning humans, Arnold et al. found an inverse relationship between circulating CD16^+^ monocytes (high) and M2 macrophages (low) that marked coronary disease severity. The differences in polarization of macrophages remained even during one week in a cell culture ex vivo. Furthermore, the authors postulate that circulating monocytes may influence the polarization of lesion macrophages, and that these measures of monocyte and macrophage subtypes may hold potential as new useful biomarkers in cardiovascular disease [43].

### 3.2. Inflammation—T-Cells (Adaptive Immune Response)

Although the majority of cells within advanced human atherosclerotic lesions are macrophages, depending on plaque morphology, around 40% of all cells represent CD3^+^ T cells. Natural Killer (NK) cells (~1%) and B cells (~2%) show minor frequencies [44]. Thus, the research focused on T cells during the last decades because they are involved in all stages of the atherosclerotic process, and act as a kind of remote control of the inflammatory process.

#### Sequential Immune Activation in AS

Endothelial injury is the first step of atherosclerosis. This way, low-density lipoprotein (LDL) reaches the intima of the vessel wall, where it undergoes enzymatic/oxidative modifications and is ingested by macrophages. This provokes foam cell transformation accompanied by the production of pro-inflammatory TNF-α, IL-1β, monocyte chemoattractant 1 (MCP-1), leukotriene B 4 (LTB4), and proteolytic metalloproteinases (MMP) [9]. Thus stimulated, endothelial cells start to overexpress adhesion molecules like VCAM-1 and ICAM-1, which fetches effectors of the adaptive immune response. Preferentially, Th1/Th17 lymphocytes begin to accumulate in the sub-endothelial space around the lipoproteins [9,45]. Macrophages and dendritic cells present antigens (e.g., Apo B 100) through MHC class II molecules to CD4^+^ T helper cells. T helper cells of type 1 (Th1) begin to release the proatherogenic cytokines IFN-γ and TNF-α. Regulatory T cells (Treg) counterbalance this activation, at least in part by the local release of TGF-β and IL-10 [9]. If this process does not abate, extra-lesional amplification loops begin to work. Antigen-loaded dendritic cells (DC) drift through lymphatic vessels to lesion draining lymph nodes and/or spleen. In lymph nodes and the spleen, naive T cells develop into effector T cells and re-enter the bloodstream [9]. When these cells arrive at the atherosclerotic lesion, they convey antibody-specific responses. Within markedly variant time, the stability of the lesion declines, the inflammation and pro-coagulant factors increase, and plaque rupture and thrombosis become more probable. Other immunologic contact barriers (e.g., bowel mucosa to gut microbiome, mucosal viral infections) may contribute to extra-lesional amplification processes [46,47]. Recently, the synthesis of trimethylamine-*N*-oxide (TMAO) from dietary phosphatidylcholine by the intestinal microbiota was brought into connection with the activity of atherosclerotic vessel disease. Increased blood and urine TMAO levels correlated positively with forthcoming cardiovascular events [48]. Viral infections (e.g., common colds, influenza) may also stimulate the atherosclerotic process via T cell activation loops. Th1 cell derived INF-γ increases the macrophage MMP-9/TIMP-1 production ratio. Thus more generated active MMP-9 destabilizes atherosclerotic lesions by increased proteolysis [9].

Taken together, T-lymphocytes are centrally involved in the atherosclerotic process. Among CD4^+^ T cells, Th1 cells are proatherogenic, Treg cells atheroprotective, and the role of Th2, Th17, and CD8^+^ T-cells remains unclear [49,50]. The relevance of T-cells for atherosclerosis has been demonstrated impressively by genetic depletion models. For example, loss of CD47, also known as integrin-associated protein, caused increased T-cell activation with increased atherosclerosis in CD47-deficient mice [51].

### 3.3. Inflammation—The Innate Immune Response (Toll-Like Receptors)

The toll-like receptors TLR2 [52], TLR4 [53], and TLR7 [54] are involved in AS and endothelial inflammation [55]. Ye Z et al. showed in humans and in ApoE-deficient mice that the P-selectin and P-selectin glycoprotein ligand axis activates dendritic cells by a toll-like receptor 4 (TLR4) signaling pathway. This contributes to the acceleration of AS [56]. TLR7 deficiency accelerated AS in ApoE-/-mice and promoted vulnerable plaque phenotypes with greater lesion size, fewer smooth muscle cells, lower collagen content, stronger infiltration of macrophages, and larger lipid deposits [54]. On the other hand, an increased TLR7 expression in human carotid plaques activated gene sequences favoring a more stable plaque phenotype. These included the M2 macrophage markers IL-10, IL-1RN, CD163, CLEC4A, CLEC7A, MSR1, CD36, MS4A4A, CLEC10A, CLEC13A/CD302, and CD209. Genes related to thrombosis, like CD40L, TF/CD142, PF4/CXCL4, vWF, GPIbα/CD42b, GPIbβ/42c, GPIIb/CD41, GPIIIa/CD61, GPV/CD42d, and GPIX/CD42a are down regulated by TLR7 [54].

### 3.4. Inflammation—B-Cells

Although the role of T-cells has been extensively studied during the last decades, the role of B-cells has recently gained more attention [57]. Mouse models showed that the depletion of B-cells attenuates plaque development. This suggests that antigen presentation of B-cells can promote atherosclerotic progression [58]. Indeed, differential effects of different B-cell subsets exist. B1 cells have been shown to prevent lesion formation, whereas B2 cells promote it [57]. Obviously, IgM antibodies from B1 cells have atheroprotective effects. The role of other immunoglobulin classes remains open. Taken together, several recent studies have established an important modulatory role of B-cells in experimental atherosclerosis. The role of immunoglobulins and the different immunoglobulin classes requires further investigation. Insights from patients with rheumatoid arthritis with increased cardiovascular risk may help in understanding of the role of B-cells, if these patients are treated with C cell–depleting drugs [57].

### 3.5. Inflammation—Plaque Erosion

Superficial intimal erosion is a common cause of thrombotic complications of AS [59]. Interestingly, mechanisms of erosion are less in focus of research than mechanisms that cause plaque rupture. However, intensive statin therapy decreases the susceptibility of plaques for rupture. Hence, erosion may contribute more to the current burden of risk. Recently, the innate immune receptor toll-like receptor 2 (TLR2) was brought into connection with altered endothelial function and superficial erosion [60]. Data from human endothelial cell cultures and atherosclerotic mice suggested overexpression of TLR2 in regions of disturbed blood flow [61]. TLR2 can bind lipoproteins (oxidized LDL) with the scavenger receptor CD36 and is involved in foam cell formation and inflammation [62]. These observations link immune-mediated inflammation and lipid peroxidation with plaque erosion. Taken together, plaque erosion has a different pathology compared with plaque rupture and thus requires distinctive therapeutic approaches. The EROSION study showed that, for patients with ACS caused by plaque erosion, a conservative treatment with anti-thrombotic therapy without stenting may be an option [63].

### 3.6. Inflammation—Matrixmetalloproteinases (MMPs)

The secretion of MMPs is not enough to account for plaque destabilization. Instead, the ratio between MMPs and their inhibitors (TIMPs) is critical. An increased MMP/TIMP ratio with high amounts of free active MMP also promotes an expansion of pro-inflammatory macrophage subsets, and extracellular matrix destruction alongside interminable monocyte/macrophage accumulation is associated with plaque rupture [64]. Notably, patients with STEMI compared to patients with stable angina pectoris had stronger elevated plasma levels of MMP12 due to an imbalance between MMP12 and TIMP1 [65]. Otherwise, beneficial effects of MMPs also exist. These include a promotion of vascular smooth muscle growth leading to plaque stabilization [64]. Taken together, members of the metalloproteinase family and their endogenous tissue inhibitors perform complex roles during late progression and rupture of atherosclerotic plaques.

### 3.7. Inflammation—Calprotectins and Danger-Associated Molecular Patterns (DAMPs)

The traditional cardiovascular risk factors, smoking, obesity, hyperglycemia, and dyslipidemia increase the DAMP proteins S100A8 and S100A9, which belong to the S100 calgranulin family. These proteins are endogenous ligands of the toll-like receptor 4 and the receptor for advanced glycation end products. Levels of S100A8 and S100A9 correlated in humans with the extent of coronary and carotid atherosclerosis and, more importantly, with a vulnerable plaque phenotype [53]. Recently, the association between plasma calprotectin and risk of cardiovascular disease was investigated in 5290 participants of the PREVEND prospective cohort study. Patients had a log-linear association between the risk of CVD and calprotectin. A limitation of this observation is the fact that concentrations of calprotectin concentration were partly dependent on high sensitive CRP. Nevertheless, to measure calprotectin in addition to conventional risk factors may improve CVD risk assessment [66].

### 3.8. Inflammation—Mast Cells

An influence of mast cells on adverse cardiovascular events is a matter of debate [67]. Mast cells accumulate in human atherosclerotic tissue, particularly in the shoulder region of the plaque [67]. Showing an increased CD63 expression a high percentage of these intraplaque mast cells are in an active stage [67]. In human carotid plaques, mast cells and tryptase content are associated positively with intraplaque vessel density and future cardiovascular events [68]. In ApoE^−/−^ mice on high triglyceride and cholesterol “Western-type” diet and restraint stress, mast cell activation correlated with plaque destabilization [69]. Mast cells of advanced human atherosclerotic lesions express Human Leucocyte Antigen (HLA)-DR, suggesting modulation of adaptive immunity by acting as antigen presenting cells [70]. Activated human mast cells release tryptase, chymase, carboxypeptidase A3, cathepsin G, and granzyme B. Of these, cathepsin G efficiently degrades apoB-100 to induce LDL fusion and to enhance the binding of LDL to human aortic proteoglycans and atherosclerotic lesions. Immunofluorescence staining of human atherosclerotic coronary arteries for tryptase and cathepsin G showed that mast cells found in these AS lesions contain cathepsin G [71]. Thus, cathepsin G from local mast cells may contribute to atherogenesis by enhancing LDL retention [71]. These observations suggest that mast cells play a role in the process of destabilization of AS lesions [72].

### 3.9. Inflammation—Biomarkers (Pentraxin 3, Myeloperoxidase, Adiponectin)

Pentraxin 3 (PTX3), a CRP-family member, is not expressed in the liver but is detectable in AS lesions. Immunochemical analysis showed that PTX3 is present in the basement membrane in endothelial and perivascular cells of carotid endarterectomy specimens from culprit atherosclerotic lesions [73]. These facts suggest PTX3 as a target candidate for molecular imaging of destabilizing processes in AS plaques [73,74,75].

Myeloperoxidase (MPO), a member of the XPO subfamily of peroxidases, produces hypochlorous acid and chloride anion during the neutrophil’s respiratory burst, which leads to oxidative damage [76]. Released by circulating phagocytes, MPO contributes to endothelial dysfunction by limiting NO bioavailability through formation of reactive oxidants. Thus, MPO is involved in the progression of atherosclerosis and may represent a potential culprit. In humans, MPO associates positively with carotid plaque inflammation [77]. Inhibition of MPO might be a new therapeutic target for AS [78], especially to limit endothelial dysfunction in vascular inflammation [79].

Osteopontin (OPN) is another biomarker of interest. Elevated serum levels predicted major adverse cardiovascular events in patients with severe carotid artery stenosis [80]. Qiao et al. suggested osteopontin-specific up-conversion nanoprobes for molecular imaging of vulnerable atherosclerotic plaques in vivo [81]. However, a recent comparative study investigating protein expression levels of five biomarkers (i.e.,: MMP-1, MMP-9, osteopontin, TNF-α, IL-6) in relation to carotid plaque classification in patients after carotid endarterectomy found higher levels of MMP-1, MMP-9, and IL-6 than OPN in plaque tissue [13]. Guo et al. investigated MMP profiles of vulnerable plaques and confirmed negative results concerning OPG [82].

Adiponectin is a protein hormone that modulates metabolic processes, especially glucose regulation and fatty acid oxidation [83]. It is secreted from adipose tissue into the bloodstream and is abundant in plasma relative to many other hormones [2]. Peripheral blood adiponectin levels were systemically lower in patients with vulnerable, as compared to patients with stable AS plaques [2,73]. We investigated in detail the role of adiponectin in obesity-associated pre-atherosclerosis of humans [84,85,86,87,88]. Moreover, we analyzed the potential of fluorescence-labeled globular adiponectin (gAd) and free adiponectin subfractions (fAd-Sfs) to bind to atherosclerotic lesions in ApoE-deficient mice [2,89,90]. These studies showed only a low binding efficiency of fAd-Sfs and a strong accumulation of gAd in the fibrous cap of AS-plaques [89,90]. These observations prompted us to discuss a role of gAd as a target sequence for the molecular imaging of AS-lesions [89]. Furthermore, we developed nano-constructs between gAd and PEGylated stealth liposomes [90], which can deliver a high number of signal-emitting molecules to AS lesions [90]. Other nano-constructs between gAd and protamine-oligonucleotide NPs, called proticles [91,92,93], displayed a particular affinity for monocytes and macrophages, which was lower when the proticles were not associated with gAd. This gAd-proticle nano-construct may be of interest for sequential AS plaque targeting. In summary, our results indicate the potential of gAd-targeted nanoparticles for the molecular imaging of AS and for possible therapeutic strategies, since it is known that gAd plays an important regenerative role [94,95,96]. However, the significance of these observations for the human situation remains to be clarified.

## 4. Balance between Clotting and Bleeding

Neovascularization is an important feature of plaque vulnerability because it stimulates vessel rupture, hemorrhage, and inflammation [97]. Intraplaque bleeding promotes the atherosclerotic destabilization process and associates with clinical endpoints. The hemoglobin/haptoglobin scavenger receptor (CD163), IL-10, hemoxygenase 1 (HO-1), ferritin, and 4-hydroxy-2-nonenal are strongly expressed in culprit lesions of patients with unstable angina pectoris [98]. Extracellular hemoglobin originating from intraplaque hemorrhage induces oxidative tissue damage by hem iron. Free hem also stimulates MPO because this molecule is a hem-containing peroxidase. The clearance mediated by a macrophage’s Hb scavenger receptors may alter the atherosclerotic process to a more aggressive one [98].

Nevertheless, the bleeding process also shows positive aspects. Induced by free hemoglobin/haptoglobin, M2 macrophages can develop into so-called M(Hb)-type macrophages, which produce anti-inflammatory cytokines, show a decreased lipid uptake, an increased cholesterol efflux, and a less foamy nature. Thus, bleeding processes within plaques do not imperatively lead to destabilization. They can also switch macrophage activities to a more stabilizing influence over time if an acute event does not interrupt this process.

## 5. Future Perspectives—Attempts to Target and Modify the Immune-Inflammatory Process in AS-Plaques

It is well established that inflammation is a central player of AS. Nevertheless, although anti-inflammatory therapy may represent a guarantor for effective treatment, only a few successful approaches exist so far to manage cardiovascular diseases with such methods. An explanation for this may be that a systemic administration of potent anti-inflammatory agents frequently generates severe side effects by misbalancing or suppressing the immune reaction. Targeted local delivery and accumulation is more promising because it can transport a drug into the “right” plaque at the right time. Typically, vulnerable and potential culprit lesions exist in the neighborhood of stable ones. Hence, it is important to detect the right candidate lesion and to place the drug in the inflammatory zone without brusque actions leading to plaque rupture or bleeding.

Recently, the Canakinumab Anti-Inflammatory Thrombosis Outcomes Study Cardiovascular Inflammation Reduction (CANTOS) trial showed that systemic administration of biologicals is promising approach [99]. In this regard targeting the interleukin-1ß pathway with canacinumab at a dose of 150 mg every three months led to a significantly lower rate of recurrent cardiovascular events than placebo. Notably, this occurred independently of lipid lowering. Alternatively, low-dose methotrexate given as anti-inflammatory agent in the Cardiovascular Inflammation Reduction (CIRT) trial failed [100] due to side effects and lacking efficacy of methotrexate. These facts show that effective therapy requires a profound basis of understanding of the immune-inflammatory reaction in atherosclerosis.

## 6. Conclusions

Cardiovascular disease is the worldwide number one killer. Hence, to diagnose a vulnerable plaque phenotype well before fatal clinical endpoints is one of the most important challenges facing personalized medicine. A comprehensive understanding of the immune-inflammatory component of the atherosclerotic process improves diagnosis and treatment of this deadly disease. Thus, the search for promising biomarker candidates in peripheral blood and in vascular lesions for molecular imaging must go on. Therapeutic modification of the inflammatory activation is definitely helpful in cases of atherosclerotic heart disease as shown recently by the CANTOS trial. Concerning stroke, no data exist so far. Undoubted, the best approach for both stroke and MI would be safe diagnosis and knock out of the vulnerable lesion before endpoints occur.

To achieve this goal, researchers must overcome the following challenges:(i)The identification of a vulnerable lesion at the right time in the right person.(ii)To act in a specific and effective way without side effects because the “patient” may still feel healthy at the time of the successful pre-diagnosis.

An intense interdisciplinary cooperation between laboratory medicine (new biomarkers), nanotechnology (nanocarriers), and radiology (molecular imaging) will pave the way for future success in saving more lives.

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
