# Peer review of "Immune-Mediated Inflammation in Vulnerable Atherosclerotic Plaques"

_molecules, 2019, doi:10.3390/molecules24173072_

Round 1

Reviewer 1 Report

This study by Mangge et al identifies modulation of immune-mediated inflammation as a new promising point of action for the eradication of fatal atherosclerotic endpoints.  Overall a massive effort by the authors to comprehensively compile the recent updates and the relevant literature in the field. 

The authors should include a section for Future perspective and include the role of potential anti-inflammatory therapies in treatment, prevention, attenuation of atherosclerosis. 

Reviewer 2 Report

Within their review article manuscript the authors Harald Mangge and Gunter Almer describe and review literature connected with atherosclerotic lesion progression and possible treatment option. The review is nicely written, however the authors try to cover a lot of topics in a rather short format. This leads to unprecise passages within the review article.

Major concerns:

Proper references for calcification within atherosclerotic lesions is missing.

Dendritic cells should be considered, especially in the role of antigen presentation.

As MMPs are secreted in a pro form, the pure secretion of MMPs is not enough to account for plaque destabilization but requires activation. Macrophage subset activation of MMPs should therefore be considered as a key factor in defining macrophage activation.

The authors do not mention the contribution of macrophage apoptosis to atherosclerotic lesion progression.

The paragraph about macrophage polarization should be rewritten. First, the M1 and M2 nomenclature should be swapped for a more generalized one. Second, the authors describe only mouse subsets, human data needs to be included.

Within the first couple of sentences of the 'Sequential immune activation in AS' paragraph literature should be cited accordingly.

The role of T-cells in atherosclerosis was already demonstrated beautifully using depletion and genetic models, this has should be added to the review.

Furthermore, even though B-cells are only present in small numbers, they are capable of modulating the local environment dramatically. Respective publications should be included.

The given inflammation biomarkers chosen seem arbitrary, a reason why only these three are mentioned should be given.

The section about microvesicles is very short and unprecise. It should be either left out or expanded (including the definition of microvesicle versus extracellular vesicle).

The authors do not mention the concept of plaque erosion.

Round 2

Reviewer 2 Report

All questions were answered.